# Research on Gradient-Descent Extended Kalman Attitude Estimation Method for Low-Cost MARG

**DOI:** 10.3390/mi13081283

**Published:** 2022-08-09

**Authors:** Ning Liu, Wenhao Qi, Zhong Su, Qunzhuo Feng, Chaojie Yuan

**Affiliations:** Beijing Key Laboratory of High Dynamic Navigation Technology, Beijing Information Science & Technological University, Beijing 100101, China

**Keywords:** gradient descent, Kalman filter, MARG, attitude estimation, data fusion

## Abstract

Aiming at the problem of the weak dynamic performance of the gradient descent method in the attitude and heading reference system, the susceptibility to the interference of accelerometers and magnetometers, and the complex calculation of the nonlinear Kalman Filter method, an extended Kalman filter suitable for a low-cost magnetic, angular rate, and gravity (MARG) sensor system is proposed. The method proposed in this paper is a combination of a two-stage gradient descent algorithm and the extended Kalman filter (GDEKF). First, the accelerometer and magnetometer are used to correct the attitude angle according to the two-stage gradient descent algorithm. The obtained attitude quaternion is combined with the gyroscope measurement value as the observation vector of EKF and the calculated attitude of the gyroscope and the bias of the gyroscope are corrected. The elimination of the bias of the gyroscope can further improve the stability of the attitude observation results. Finally, the MARG sensor system was designed for mathematical model simulation and hardware-in-the-loop simulation to verify the performance of the filter. The results show that compared with the gradient descent method, it has better anti-interference performance and dynamic performance, and better measurement accuracy than the extended Kalman filter.

## 1. Introduction

Navigation and guidance technologies are widely used in industrial and military fields such as unmanned aerial vehicles (UAV), autonomous underwater vehicles, mobile devices, and human motion tracking. Accurate attitude measurement is essential for the development of navigation and guidance technologies [1,2,3]. If a single type of sensor is used to measure the attitude of the carrier, different sensors have their own weak points [4,5]. For example, the accelerometer cannot separate the gravitational acceleration and the linear acceleration generated during the movement of the carrier, while the gyroscope is easily affected by temperature drift and noise signals, resulting in excessive accumulated errors, and the output of the magnetometer is easily interfered by magnetic materials near the sensor. In recent years, MARG (magnetic, angular rate, and gravity) sensor systems have received widespread attention [6]. Therefore, data from different sensors need to be integrated to provide an accurate position and attitude estimation [7,8].

Among the existing inertial and magnetic sensor attitude estimation methods, the most commonly used method is the complementary filtering method. Mahony et al., proposed a complementary filtering algorithm for UAV attitude calculation [9]. Then, Liang et al., applied the complementary filtering method to the combination of inertia and magnetometer for attitude calculation [10]. Calusdian et al., proposed a quaternion-based adaptive gain complementary filter [11]. This complementary filtering algorithm uses a proportional integral (PI) controller to estimate the gyroscope deviation and provide a decent attitude estimation. Madgwick et al., proposed a gradient descent method for human body motion posture tracking [12], which can reduce the influence of magnetic interference. In addition, with the development of satellite technology, the measurement method combining an inertial measurement unit and global navigation satellite system (GNSS) can improve the accuracy of vehicle attitude measurement under actual driving conditions [13]. These methods do not involve the processing of system errors and sensor measurement noise.

With the increasing requirements for system stability and reliability, it is necessary to control the internal data noise of the system [14,15]. Since the Kalman filter is a method derived for the purpose of minimizing the mean square error in the process of estimating the state of the linear system, it has been widely used [16,17], especially in the field of attitude calculation [18]. Sabatini et al., proposed a quaternion-based extended Kalman filter for human body attitude tracking [19]. The accelerometer and magnetometer were used as state vectors to construct the Kalman filter. In Ref. [20], the magnetic disturbance and gyroscope bias error are modeled as the state vector of the filter, and the measurement model is linearized by calculating the Jacobian matrix. However, the standard extended Kalman filter must linearize the process model or measurement model, which inevitably introduces linearization errors. In addition, the extended Kalman linearization process increases the amount of calculation of the microcontroller.

In order to avoid the linearization of the measurement model and reduce the calculation load based on the quaternion extended Kalman filter, a two-layer filter architecture is proposed in [21,22]. In the first layer, a quaternion estimation (QUEST) is designed, which estimates the attitude quaternion according to the measured values of the accelerometer and magnetometer to solve the Wahba problem [23]. In the second layer, a linear Kalman filter is designed, and the quaternion calculated in the first layer is used as the observation vector, which avoids the linearization error introduced by the linearization process of the observation model and simplifies the design of the Kalman filter. Marins et al., used the Gauss–Newton iterative algorithm to find the optimal quaternion estimate through the accelerometer and magnetometer [21]. Liu et al., obtained the optimal weight of each measurement value by calculating the error variance and proposed an improved quaternion Gauss–Newton method for attitude estimation [22]. Different from the idea of designing a quaternion estimator (QUEST), in [24,25], the factored quaternion algorithm (FQA) [26] is used to solve the quaternion, and then input to the Kalman filter as the observation vector. However, it can be obtained from the error analysis of [27] that when the carrier is in a fast movement, the above algorithm has a large error.

In this article, in order to solve the Whaba problem, a two-stage gradient descent algorithm optimization method is adopted, which uses the accelerometer and magnetometer to calculate the quaternion and provides the measurement value for the extended Kalman filter. In the two-stage gradient descent algorithm, the first stage uses the accelerometer model to construct the error function, and the gradient descent algorithm is used to update the carrier pitch and roll angle. The second stage uses a gradient descent algorithm to update the yaw angle of the carrier based on the magnetometer measurement model. At the same time, a simple adaptive step size method is proposed, which has better measurement accuracy compared to the factored quaternion algorithm (FQA) in dynamic conditions. The quaternion obtained by the two-stage update and gyroscope measurement value is used as the observation value, and as the input of the extended Kalman filter. In the designed Kalman filter, the gyroscope output quaternion and bias of gyroscope is used as the predicted value, and the optimal estimate of the final prediction is updated by combining the predicted value with the new measured value. Finally, compared with the extended Kalman (EKF), gradient-descent linear Kalman filter (GDLKF) and gradient-descent (GD) algorithm, the proposed method has stronger robustness, and has a certain ability to resist magnetic interference.

The structure of this article is as follows. In Section 2, the attitude representation method is briefly introduced. Through the establishment of accelerometer and magnetometer models, the basic principles of attitude determination are introduced. In Section 3, the specific ideas of the proposed extended Kalman filter and the design methods of different processes are introduced. The experimental results are discussed in Section 4. Section 5 is the conclusion.

## 2. Attitude Representation and Determination

### 2.1. Coordinate System Establishment

In order to express the attitude, the two coordinate systems shown in Figure 1 are used to represent the carrier coordinate system $b\left\{xyz\right\}$ and the navigation coordinate system n. Among them, the carrier coordinate system adopts the front-right-down coordinate system that conforms to the right-hand rule: the *x*-axis points to the front of the carrier, the *y*-axis points to the right side of the carrier, and the *z*-axis points to the carrier vertically downward. The navigation coordinate system adopts the general North East Down (NED) coordinate system.

### 2.2. Attitude Representation

In the three-dimensional space, the Euler angle representation method is used to describe the transformation from one coordinate system to another coordinate system through three consecutive rotations around different coordinate axes. According to the definition of the Euler angle, rotate around z-axis, y-axis and x-axis in turn to obtain yaw angle ψ, pitch angle θ and roll angle ϕ. Three rotations can be mathematically expressed as an independent direction cosine matrix, as shown in Equation (1). Then, the direction cosine matrix $C_{n}^{b}$ that transformed the carrier coordinate system to the reference coordinate system can be derived as shown in Equation (2).
(1)$$C_{\psi }^{z}=\left[\begin{array}{ccc} \cos\psi & -\sin\psi & 0\\ \sin\psi & \cos\psi & 0\\ 0 & 0 & 1 \end{array}\right],C_{\theta }^{y}=\left[\begin{array}{ccc} \cos\theta & 0 & \sin\theta \\ 0 & 1 & 0\\ -\sin\theta & 0 & \cos\theta \end{array}\right],C_{\phi }^{x}=\left[\begin{array}{ccc} 1 & 0 & 0\\ 0 & \cos\phi & -\sin\phi \\ 0 & \sin\phi & \cos\phi \end{array}\right]$$
(2)$$C_{n}^{b}=C_{\psi }^{z}C_{\theta }^{y}C_{\phi }^{x}=\left[\begin{array}{ccc} \cos\theta \cos\psi & \cos\theta \sin\psi & -\sin\theta \\ \sin\phi \sin\theta \cos\psi -\cos\phi \sin\psi & \sin\phi \sin\theta \sin\psi +\cos\theta \cos\psi & \sin\phi \cos\theta \\ \cos\phi \sin\theta \cos\psi +\sin\phi \sin\psi & \cos\phi \sin\theta \sin\psi -\sin\phi \cos\psi & \cos\phi \cos\theta \end{array}\right]$$

However, due to the singularity problem in the attitude calculation of the above method, this paper adopts the quaternion method to express the attitude of the carrier. The transformation from one coordinate system to another coordinate system can be achieved by making a single rotation angle α around a vector r defined in the reference coordinate system. The quaternion q is defined as:(3)$$q=\left[\begin{array}{cccc} q_{0} & q_{1} & q_{2} & q_{3} \end{array}\right]=\left[\begin{array}{cccc} \cos\frac{\alpha }{2} & -r_{x}\sin\frac{\alpha }{2} & -r_{y}\sin\frac{\alpha }{2} & -r_{z}\sin\frac{\alpha }{2} \end{array}\right]$$

Unlike the complex rotation matrix derived from Equation (2), the directional cosine matrix of the quaternion q can be simply described as:(4)$$C_{n}^{b}=\left[\begin{array}{ccc} q_{0}^{2}+q_{1}^{2}-q_{2}^{2}-q_{3}^{2} & 2(q_{1}q_{2}-q_{0}q_{3}) & 2(q_{1}q_{3}+q_{0}q_{2})\\ 2(q_{1}q_{2}+q_{0}q_{3}) & q_{0}^{2}-q_{1}^{2}+q_{2}^{2}-q_{3}^{2} & 2(q_{2}q_{3}-q_{0}q_{1})\\ 2(q_{1}q_{3}-q_{0}q_{2}) & 2(q_{2}q_{3}+q_{0}q_{1}) & q_{0}^{2}-q_{1}^{2}-q_{2}^{2}+q_{3}^{2} \end{array}\right]$$

The change angle of the Euler angle ψ, ϕ and θ of the carrier can be obtained by the direction cosine matrix decomposition [28]:(5)$$\left[\begin{array}{c} \phi \\ \theta \\ \psi \end{array}\right]=\left[\begin{array}{c} \arctan2(q_{0}q_{1}+q_{2}q_{3})/(q_{0}^{2}-q_{1}^{2}-q_{2}^{2}+q_{3}^{2})\\ -\arcsin2(q_{1}q_{3}-q_{0}q_{2})\\ \arctan2(q_{1}q_{2}+q_{0}q_{3})/(q_{0}^{2}+q_{1}^{2}-q_{2}^{2}+q_{3}^{2}) \end{array}\right]$$

### 2.3. Attitude Determination

The accelerometer can determine the pitch and roll attitude of the carrier by measuring the acceleration of gravity under static conditions. According to the pitch and roll attitude information provided by the accelerometer, the magnetometer can determine the yaw attitude information of the carrier by measuring the geomagnetic field of the environment without magnetic interference, and then obtain the overall attitude information of the carrier.

#### 2.3.1. Determination of Pitch Angle and Roll Angle

Ideally, the carrier in the static state is only affected by the acceleration of gravity g. The measured output of the accelerometer in the carrier coordinate system can be represented as ab=axayazT. Then, substitute the direction cosine matrix of Equation (2) to obtain the result:(6)$$\left[\begin{array}{c} a_{x}\\ a_{y}\\ a_{z} \end{array}\right]=C_{n}^{b}\left[\begin{array}{c} 0\\ 0\\ g \end{array}\right]=\left[\begin{array}{c} -g\sin\theta \\ g\sin\phi \cos\theta \\ g\cos\phi \cos\theta \end{array}\right]$$

Moreover, the pitch angle and roll angle can be derived:(7)$$\theta =\arcsin(\frac{a_{x}}{g})$$
(8)$$\phi =\arctan(\frac{-a_{y}}{a_{z}})$$

#### 2.3.2. Determination of Yaw Angle

Since the accelerometer can only measure the angle of the carrier relative to the horizontal plane, in order to obtain the heading information of the carrier, it is necessary to use a magnetometer to measure the north component of the geomagnetism at the position of the carrier, assuming that the output of the geomagnetic field measured by the magnetometer in the carrier coordinate system is hb=hxbhybhzbT, and the horizontal component of the geomagnetic field vector is expressed as hl=hxlhylhzlT. According to the the Equations (7) and (8), pitch and roll angles derived from the direction cosine matrix in Equation (2), hl can be calculated:(9)hxlhylhzl=cosθsinθ0−cosψsinθcosϕcosθsinϕsinθsinϕ−sinϕcosθcosϕhxbhybhzb

Then, the yaw angle can be obtained [29]:(10)ψ=arctan(hzlhxl)+D

Among them, D represents the magnetic declination, the angle between the north direction of the geomagnetic field and the north direction of the navigation coordinate system, which varies with the location of the carrier.

## 3. Gradient Descent Kalman Filter Algorithm

This paper proposes an extended Kalman filter data fusion method. The overall block diagram is shown in Figure 2. The gyroscope is used as the state vector of the system state and input into the process model of the filter. Thus, the predicted value $X_{k}$ is obtained. The measurement model of the filter uses the gradient descent algorithm, which takes the measured values of the accelerometer and the magnetometer and gyroscope as input. The attitude quaternion is used as the observation value of the extended Kalman filter to update the predicted value calculated by the gyroscope.

### 3.1. Process Model

In the process model, the Kalman filter proposed in this paper selects the state vector composed of the quaternion form $q_{\omega }$ of the three-axis angular rate output by the gyroscope and angular velocity bias $\omega_{b}$. The state vector is represented by Equation (11).
(11)$$X_{k}={\left[\begin{array}{cc} q_{\omega } & \omega_{b} \end{array}\right]}^{T}={\left[\begin{array}{lllllll} q_{\omega ,0} & q_{\omega ,1} & q_{\omega ,2} & q_{\omega ,3} & \omega_{xb} & \omega_{yb} & \omega_{zb} \end{array}\right]}^{T}$$

#### 3.1.1. State Prediction

We assume that the angular velocity bias does not change very much from one sample to the next. When calculating the prediction of the attitude quaternion in discrete time, the derivative of the quaternion needs to be numerically integrated, as shown in the equation.
(12)$$q_{k}=q_{k-1}+\Delta t\cdot {\dot{q}}_{\omega ,k}$$

The angular rate measured by the gyroscope obeys a vector differential equation, which describes the attitude change rate as a quaternion derivative [30]:(13)$${\dot{q}}_{\omega }=\frac{1}{2}\omega_{q,k}q_{\omega ,k}=\frac{1}{2}\left[\begin{array}{lll} -q_{1} & -q_{2} & -q_{3}\\ q_{0} & q_{3} & -q_{2}\\ -q_{3} & q_{0} & q_{1}\\ q_{2} & -q_{1} & q_{0} \end{array}\right]\left[\begin{array}{c} \omega_{x}\\ \omega_{y}\\ \omega_{z} \end{array}\right]$$

Therefore, the attitude quaternion qω,tb of the state space equation arises as in Equation (14).
(14)$$f(q_{k})=F\left(k\right)X\left(k\right)=\left[\begin{array}{c} q_{0}+\frac{1}{2}\Delta t(q_{1}\omega_{x}+q_{2}\omega_{y}+q_{3}\omega_{z})\\ q_{1}+\frac{1}{2}\Delta t(q_{0}\omega_{x}+q_{3}\omega_{y}-q_{2}\omega_{z})\\ q_{2}+\frac{1}{2}\Delta t(q_{3}\omega_{x}-q_{0}\omega_{y}-q_{1}\omega_{z})\\ q_{3}+\frac{1}{2}\Delta t(q_{2}\omega_{x}-q_{1}\omega_{y}+q_{0}\omega_{z}) \end{array}\right]$$

Taking the Jacobian matrix of the state space equation, the state transition matrix of Equation (15) is obtained.
(15)$$\Phi_{k}=\frac{\partial f(X_{k})}{\partial X_{k}}=\left[\begin{array}{ccccccc} 1 & -\frac{\Delta t}{2}\omega_{x} & -\frac{\Delta t}{2}\omega_{y} & -\frac{\Delta t}{2}\omega_{z} & 0 & 0 & 0\\ \frac{\Delta t}{2}\omega_{x} & 1 & \frac{\Delta t}{2}\omega_{z} & \frac{\Delta t}{2}\omega_{y} & 0 & 0 & 0\\ \frac{\Delta t}{2}\omega_{y} & \frac{\Delta t}{2}\omega_{z} & 1 & -\frac{\Delta t}{2}\omega_{x} & 0 & 0 & 0\\ \frac{\Delta t}{2}\omega_{z} & -\frac{\Delta t}{2}\omega_{y} & \frac{\Delta t}{2}\omega_{x} & 1 & 0 & 0 & 0\\ 0 & 0 & 0 & 0 & 1 & 0 & 0\\ 0 & 0 & 0 & 0 & 0 & 1 & 0\\ 0 & 0 & 0 & 0 & 0 & 0 & 1 \end{array}\right]$$

#### 3.1.2. System Noise

The system noise of the prediction process mainly comes from the gyroscope. The part of quaternion noise can be expressed by Equation (16):(16)$$\left[\begin{array}{c} {\dot{q}}_{0}\\ {\dot{q}}_{1}\\ {\dot{q}}_{2}\\ {\dot{q}}_{3} \end{array}\right]=\frac{1}{2}\left[\begin{array}{cccc} 0 & -\omega_{x} & -\omega_{y} & -\omega_{z}\\ \omega_{x} & 0 & \omega_{z} & -\omega_{y}\\ \omega_{y} & -\omega_{z} & 0 & \omega_{x}\\ \omega_{z} & \omega_{y} & -\omega_{x} & 0 \end{array}\right]\left[\begin{array}{c} q_{0}\\ q_{1}\\ q_{2}\\ q_{3} \end{array}\right]$$

The measurement output of the gyroscope $\omega ={\left[\omega_{x}\omega_{y}\omega_{z}\right]}^{T}$ mainly includes two components: the ideal value $\bar{\omega }={\left[{\bar{\omega }}_{x}{\bar{\omega }}_{y}{\bar{\omega }}_{z}\right]}^{T}$ and the drift value $\delta \omega ={\left[\delta \omega_{x}\delta \omega_{y}\delta \omega_{z}\right]}^{T}$ of the gyroscope in the sensor coordinate system, that is $\omega =\bar{\omega }+\delta \omega$. Therefore, the state equation can be rewritten as:
(17)$$\left[\begin{array}{c} {\dot{q}}_{0}\\ {\dot{q}}_{1}\\ {\dot{q}}_{2}\\ {\dot{q}}_{3} \end{array}\right]=\frac{1}{2}\left[\begin{array}{cccc} 0 & -{\bar{\omega }}_{x} & -{\bar{\omega }}_{y} & -{\bar{\omega }}_{z}\\ {\bar{\omega }}_{x} & 0 & \omega_{z} & -{\bar{\omega }}_{y}\\ {\bar{\omega }}_{y} & -{\bar{\omega }}_{z} & 0 & {\bar{\omega }}_{x}\\ {\bar{\omega }}_{z} & {\bar{\omega }}_{y} & -{\bar{\omega }}_{x} & 0 \end{array}\right]\left[\begin{array}{c} q_{0}\\ q_{1}\\ q_{2}\\ q_{3} \end{array}\right]+\frac{1}{2}\left[\begin{array}{ccc} q_{1} & q_{2} & q_{3}\\ -q_{0} & q_{3} & -q_{2}\\ -q_{3} & -q_{0} & q_{1}\\ q_{2} & -q_{1} & -q_{0} \end{array}\right]\left[\begin{array}{c} \delta \omega_{x}\\ \delta \omega_{y}\\ \delta \omega_{z} \end{array}\right]$$

In a discrete system, extract the system noise $w_{k}$ from the formula as follows:
(18)$$w_{k}=\frac{\Delta T}{2}G_{k}v_{gk}=\frac{\Delta T}{2}\left[\begin{array}{ccc} q_{0} & q_{1} & q_{2}\\ -q_{0} & -q_{3} & -q_{2}\\ q_{2} & -q_{0} & -q_{1}\\ -q_{2} & q_{1} & q_{0} \end{array}\right]\left[\begin{array}{c} \delta \omega_{x}\\ \delta \omega_{y}\\ \delta \omega_{z} \end{array}\right]$$
where ΔT is the sampling time, $v_{gk}$ is the Gaussian white noise with a mean value of zero and is normally distributed, and its covariance matrix is $\sum_{g}=\delta_{3\times 3}^{2}$, and the covariance matrix of quaternion noise $Q_{\omega }$ can be expressed as:(19)$$Q_{\omega }=E(w_{k}w_{k}{}^{T})={\left(\frac{\Delta T}{2}\right)}^{2}G_{k}\sum_{g}G_{k}{}^{T}$$

The covariance matrix of the process system noise is constructed:(20)$$Q_{k}=\left[\begin{array}{cc} Q_{\omega } & 0\\ 0 & I_{3} \end{array}\right]$$

### 3.2. Measurement Model

In this paper, the observation vector is calculated by three sensors: the accelerometer and magnetometer. The attitude quaternion calculated by the accelerometer and magnetometer as the observation value of the Kalman filter system can be expressed as $q_{\nabla }={\left[\begin{array}{cccc} q_{\nabla ,0} & q_{\nabla ,1} & q_{\nabla ,2} & q_{\nabla ,3} \end{array}\right]}^{T}$. This gives the measurement vector as shown in Equation (21).
(21)$$Z(k)={\left[\begin{array}{cc} q_{\nabla } & \omega \end{array}\right]}^{T}={\left[\begin{array}{ccccccc} q_{\nabla ,0} & q_{\nabla ,1} & q_{\nabla ,2} & q_{\nabla ,3} & \omega_{x} & \omega_{y} & \omega_{z} \end{array}\right]}^{T}$$

#### 3.2.1. Gradient Attitude Quaternion

Before inputting the data into the Kalman filter based on quaternion, it is very important to filter out the noise from the measurement process of the sensor. In this paper, the two-stage gradient descent algorithm is used. The output of the accelerometer and the magnetometer are compared with the horizontal components of the gravitational field and the geomagnetic field, respectively, to calculate the optimal attitude quaternion as the system measurement value. Figure 3 shows the schematic diagram of the gradient descent algorithm.

When the carrier is in a static state, the measured values of the accelerometer and magnetometer in the carrier coordinate system are a^b and m^b, respectively, which are converted into quaternion forms as follows:(22)a^b=0axayaz
(23)m^b=0mxmymz

In an ideal state, the accelerometer output should be the same as the gravity vector g^n in the navigation coordinate system after the direction cosine matrix conversion from the carrier coordinate system to the navigation coordinate system. In the navigation coordinate system, the gravity vector g^n can be normalized as
(24)g^n=0001

According to the measured value of the accelerometer and the reference value of the gravity vector, the target error function fg(qnb,g^n,a^b) is constructed:(25)fg(q^nb,g^n,a^b)=q^nb*⊗g^n⊗q^nb−a^b=2(q1q3−q0q2)−ax2(q1q3−q0q2)−ay2(12−q12−q22)−ax

In the gradient descent algorithm, in order to find the extreme value of the target error function, the Jacobian matrix Jg(q^nb,g^n,a^b) of the error function needs to be calculated:(26)Jg(q^nb,g^n,a^b)=−2q22q3−2q02q12q12q02q32q20−4q1−4q20

From the error function and the corresponding Jacobian matrix, the gradient of the error function ∇f(q^nb,d^n,a^b) can be obtained as
(27)∇f(q^nb,d^n,a^b)=Jg(q^nb,g^n,a^b)fg(q^nb,g^n,a^b)

The accelerometer alone cannot accurately measure the carrier’s yaw angle attitude, because it cannot sense the rotational movement in the z axis. Therefore, a magnetometer is needed for further compensation. Suppose the reference vector b^n of the magnetic field at the position of the carrier in the navigation coordinate system is:(28)b^n=0bx0bz

Assuming that after m^b is transformed from the carrier coordinate system to the reference coordinate system through the rotation matrix, the output of the magnetometer in the navigation coordinate system h^n is obtained as:(29)h^n=0hxhyhz

In the navigation coordinate system, the projection of h^n and b^n on the plane xOy should be equal, so $b_{x}^{2}=h_{x}^{2}+h_{y}^{2}$, $b_{z}=h_{z}$, that:(30)b^n=0hx2+hy20hz

At the same time, the error function fm(qnb,b^n,m^b) and corresponding Jacobian matrix Jm(qnb,b^n,m^b) can be obtained:(31)fm(qnb,b^n,m^b)=2bx(0.5−q22−q32)2bx(q1q2−q0q3)2bx(q0q2−q1q3)+2bz(q1q3−q0q2)−mx+2bz(q0q1−q2q3)−my+2bz(0.5−q12−q22)−my
(32)Jm(qnb,b^n,m^b)=−2bzq22bzq3−2bxq3+2bzq12bxq2+2bzq02bxq22bxq3−4bzq1−4bxq2−2bzq0−4bxq3+2bzq12bxq2+2bzq3−2bxq0+2bzq22bxq0−4bzq22bxq1

Combining the Equations (25) and (31) with the corresponding Jacobian matrix Equations (26) and (32), the combined error function $f_{\nabla }$ and the corresponding Jacobian matrix $J_{\nabla }$ can be obtained, respectively, as below:(33)f∇=fg(q^nb,g^n,a^s)fm(q^nb,b^n,m^s)
(34)J∇=Jg(q^nb,g^n,a^s)Jm(q^nb,b^n,m^s)

By definition, the overall gradient of the combined error function $\nabla f_{\nabla }=J_{\nabla }^{T}f_{\nabla }$ can be obtained from above.

Finally, the gradient descent algorithm is used to calculate the attitude quaternion. Bring it into the observation vector $y_{k+1}$ of the system to obtain the measurement equation of the Kalman filter:(35)$$q_{\nabla /k+1}=q_{\nabla /k}-\mu \frac{\nabla f_{\nabla }}{\left\|\nabla f_{\nabla }\right\|}$$

Among them, $q_{\nabla /k+1}$ is the optimal pose quaternion estimated by the gradient descent method and $q_{\nabla /k}$ is the optimal posture estimation value calculated last time by the proposed Kalman filter.

The general form of the gradient descent algorithm of the accelerometer and magnetometer is used in Equations (33)–(35). Since the gradient descent algorithm is a first-order iterative algorithm, in order to improve the calculation accuracy, the second derivative of Equation (33) can be used. However, this solution increases the calculation load of the overall system, which is not considered in practical applications. An alternative method is proposed here to find the best estimate of the step size μ, so that the algorithm convergence rate is greater than the carrier motion. For this reason, the step size μ is positively correlated with the system sampling time ΔT, the angular rate of the carrier motion measured by the gyroscope ωb, and the scale factor β [31]:(36)μ∼βωbΔT

Among them, β is the gain coefficient estimated according to the zero-mean measurement error of the screw instrument, and the angular velocity ω of the carrier movement can be calculated using the following equation:(37)$$\omega =\sqrt{\omega_{x}^{2}+\omega_{y}^{2}+\omega_{z}^{2}}$$

In Equation (36), when the initial state of the carrier is stationary or slowly moving, μ should take a relatively small initial value $\mu_{0}$. Finally, the adaptive step size μ can be given by the following equation.
(38)μ=μ0+βωbΔT

Among them, $\mu_{0}$ is the initial step size. The ideal values of the parameters $\mu_{0}$ and β should enable the carrier to remain stable during static testing, and to keep fast tracking during dynamic testing without excessive overshoot. The determination of these two parameters is given in the experimental section.

#### 3.2.2. Measurement Transfer

In order to obtain the measurements to align with the states the connection between measurements and states must be made. This is achieved by finding the non-linear measurement vector equation h(x) and its Jacobian. Therefore, the nonlinear equation is shown in Equation (39), where the gradient quaternion is calculated by substituting Equations (33) and (34) into Equation (35).
(39)$$h\left(x\right)=\left[\begin{array}{c} q_{0}-2\mu \left(\begin{array}{cc} \; & 2q_{0}-m_{z}b_{z}q_{0}-m_{y}b_{z}q_{1}+m_{x}b_{z}q_{2}-m_{x}b_{x}q_{0}\\ \; & -m_{z}b_{x}q_{2}+m_{y}b_{x}q_{3}-a_{z}q_{0}-a_{y}q_{1}+a_{x}q_{2} \end{array}\right)\\ q_{1}-2\mu \left(\begin{array}{cc} \; & 2q_{1}-m_{y}b_{z}q_{0}+m_{z}b_{z}q_{1}-m_{x}b_{z}q_{3}-m_{x}b_{x}q_{1}\\ \; & -m_{y}b_{x}q_{2}-m_{z}b_{x}q_{3}-a_{y}q_{0}+a_{z}q_{1}-a_{x}q_{3} \end{array}\right)\\ q_{2}-2\mu \left(\begin{array}{cc} \; & 2q_{2}+m_{x}b_{z}q_{0}+m_{z}b_{z}q_{2}-m_{y}b_{z}q_{3}-m_{z}b_{x}q_{0}\\ \; & -m_{y}b_{x}q_{1}+m_{x}b_{x}q_{2}+a_{x}q_{0}+a_{z}q_{2}-a_{y}q_{3} \end{array}\right)\\ q_{3}-2\mu \left(\begin{array}{cc} \; & 2q_{3}-m_{x}b_{z}q_{1}-m_{y}b_{z}q_{2}-m_{z}b_{z}q_{3}+m_{y}b_{x}q_{0}\\ \; & -m_{z}b_{x}q_{1}+m_{x}b_{x}q_{3}-a_{x}q_{1}-a_{y}q_{2}-a_{z}q_{3} \end{array}\right)\\ \omega_{x}+\omega_{xb}\\ \omega_{y}+\omega_{yb}\\ \omega_{z}+\omega_{zb} \end{array}\right]$$

The covariance matrix $H_{k}$ is now built by the matrices, as shown in Equation (40). This matrix is the same as calculating the Jacobian of Equation (39).
(40)$$H_{k}={\left[\begin{array}{cc} H_{\nabla } & 0\\ 0 & I_{3\times 3} \end{array}\right]}_{7\times 9}$$
where $H_{\nabla }$ is given by Equation (41).
(41)$$H_{\nabla }={\left[\begin{array}{cccccc} {}^{\partial q_{0}}/{}_{\partial a_{x}} & {}^{\partial q_{0}}/{}_{\partial a_{y}} & {}^{\partial q_{0}}/{}_{\partial a_{z}} & {}^{\partial q_{0}}/{}_{\partial m_{x}} & {}^{\partial q_{0}}/{}_{\partial m_{y}} & {}^{\partial q_{0}}/{}_{\partial m_{z}}\\ {}^{\partial q_{1}}/{}_{\partial a_{x}} & {}^{\partial q_{1}}/{}_{\partial a_{y}} & {}^{\partial q_{1}}/{}_{\partial a_{z}} & {}^{\partial q_{1}}/{}_{\partial m_{x}} & {}^{\partial q_{1}}/{}_{\partial m_{y}} & {}^{\partial q_{1}}/{}_{\partial m_{z}}\\ {}^{\partial q_{2}}/{}_{\partial a_{x}} & {}^{\partial q_{2}}/{}_{\partial a_{y}} & {}^{\partial q_{2}}/{}_{\partial a_{z}} & {}^{\partial q_{2}}/{}_{\partial m_{x}} & {}^{\partial q_{2}}/{}_{\partial m_{y}} & {}^{\partial q_{2}}/{}_{\partial m_{z}}\\ {}^{\partial q_{3}}/{}_{\partial a_{x}} & {}^{\partial q_{3}}/{}_{\partial a_{y}} & {}^{\partial q_{3}}/{}_{\partial a_{z}} & {}^{\partial q_{3}}/{}_{\partial m_{x}} & {}^{\partial q_{3}}/{}_{\partial m_{y}} & {}^{\partial q_{3}}/{}_{\partial m_{z}} \end{array}\right]}_{4\times 6}$$

#### 3.2.3. Observation Noise

Firstly, define the vector composed of accelerometer, magnetometer and gyroscope:
(42)$$u={\left[\begin{array}{ccccccccc} a_{x} & a_{y} & a_{z} & m_{x} & m_{y} & m_{z} & \omega_{x} & \omega_{y} & \omega_{z} \end{array}\right]}^{T}$$

The measurement covariance matrix of vector can be shown in Equation (43).
(43)$$\sum_{u}=\left[\begin{array}{ccc} \sum_{acc} & & \\ & \sum_{mag} & \\ & & \sum_{gyro} \end{array}\right]$$
where $\sum_{acc}$, $\sum_{mag}$ and $\sum_{gyro}$ represent the variance matrix of accelerometer, magnetometer and gyroscope, respectively.

The corresponding covariance matrix of measurement system noise is constructed:(44)$$R=H_{k}\sum_{u}H_{k}^{T}$$
where $\sum_{u}$ is the measurement covariance matrix given by Equation (43), $H_{k}$ is the observation covariance matrix given by Equation (40) and $H_{k}^{T}$ represents the transpose of $H_{k}$.

### 3.3. Kalman Filter Design

The initial conditions for the calculation of the Kalman filter proposed in this paper are:(45)$${\hat{X}}_{0}=E[X_{0}]$$
(46)$$\Phi_{0}=E[(X_{0}-X_{0}){\left(X_{0}-X_{0}\right)}^{T}]$$

The initial attitude angle information of the initial state vector quaternion ${\hat{\Phi }}_{0}$ can be calculated according to Equations (7)–(10). $\Phi_{0}$ is the initial covariance matrix. In order to enable the stability of the filter, $P_{0}$ should be given a large positive value [32], $\Phi_{0}=10I_{7\times 7}$. $I_{7\times 7}$ is a seven-dimensional identity matrix.

Next, the system state prediction is performed, and the state equation and covariance matrix are updated from time k to k+1 time:(47)$$X_{k+1/k}=F_{k+1/k}X_{k}$$
(48)$$P_{k+1/k}=\Phi_{k}P_{k}\Phi_{k}^{T}+Q_{k}$$

$Q_{k}$ is the covariance matrix of process noise, which is calculated by Equation (20).

Then, calculate the Kalman gain of the filter as follows:(49)$$K_{k+1}=P_{k+1/k}H_{k+1/k}^{T}{\left(H_{k+1/k}P_{k+1/k}H_{k+1/k}^{T}+R_{k}\right)}^{-1}$$

Among them, $R_{k+1}$ is the covariance matrix of the measurement noise, which can be determined by Equation (44).

The last step is to compare the measured value at $t_{k+1}$ with the predicted value of the measured value from the system model. According to the above algorithm, the predicted value is updated with the measured value to obtain an optimal estimate; the optimal estimation of the state variable at $t_{k+1}$ is as follows:(50)$$y_{k+1}=y_{k+1/k}+K_{k+1}(Z_{k}-H_{k+1}X_{k+1/k})$$

The covariance is:(51)$$P_{k+1}=P_{k+1/k}-K_{k+1}H_{k+1}P_{k+1/k}$$

In this way, each new measurement value collected by the system can use Equations (49)–(51) to update the system state.

## 4. Experiments and Results

### 4.1. Hardware Design

In order to verify the effectiveness and accuracy of the proposed attitude measurement method, a measurement device containing MARG sensor is designed. The measurement device integrates four LSM9DS1 modules and four ICM-42688-P modules. LSM9DS1 has three-axis digital linear acceleration sensors, a three-axis digital angular velocity sensor and a three-axis digital magnetometer. ICM-42688-P also has a six-axis inertial measurement function, which together form the MARG sensor system. The MARG sensor system can be driven in different ways, which can fully meet the needs of the measurement device for the adaptability and stability of different environments. In addition, the measuring device integrates an STM32H753 microprocessor for data acquisition, transmission and calculation. At the same time, data collection is sent to the PC through the RS-422 communication serial port. MATLAB and HDNT Center can be used for the data analysis and posture calculation. The overall size of the measuring device is about 45 mm ∗ 40 mm ∗ 20 mm, which can be widely used in various environments. The block diagram of the hardware design is shown in Figure 4 and Figure 5 and shows a picture of the designed measuring device.

### 4.2. Experiment Design and Result Analysis

The experiment is divided into a three-parts simulation experiment, hardware-in-the-loop simulation experiment and anti-interference experiment to verify the stability and performance of the system under different conditions. In the proposed method, the default value of the initial step size $\mu_{0}$ and the proportional variable β in Equation (38) is $\mu_{0}=0.01$, β=10. The zero bias of the self-designed MARG sensor system and random error standard deviation are shown in Table 1, which can be used to calculate the system noise and measurement noise of the proposed Kalman filter. The parameters of the proposed Kalman filter are shown in Equation (52).
(52)$$\begin{array}{l} Q_{k}=diag\left(\begin{array}{ccccccc} 1e^{-4} & 1e^{-4} & 1e^{-4} & 1e^{-4} & 0.1 & 0.1 & 0.1 \end{array}\right)\\ R_{k}=diag\left(\begin{array}{ccccccc} 1e^{-3} & 1e^{-3} & 1e^{-3} & 1e^{-3} & 1e^{-2} & 1e^{-2} & 1e^{-2} \end{array}\right) \end{array}$$

#### 4.2.1. Simulation Experiment

In order to verify the reliability and stability of the designed attitude calculation method, a typical sine motion model is used as the input for the simulation analysis to verify the accuracy and precision of the proposed attitude calculation method. Taking a sine motion model with a frequency of 0.2 Hz and an amplitude of 10° as the input, the result of the attitude angle solution is shown in Figure 6, Figure 7 and Figure 8. The left side is the comparison diagram between the Euler angle solution result and the input motion model, and the right side shows the error between the solution result and the true value. It can be seen from the figure that the proposed attitude calculation method has a maximum error of about 0.4° for the roll angle and pitch angle, and a maximum error of about 0.5° for the yaw angle.

#### 4.2.2. Static Hardware-in-the-Loop Simulation Experiment

The vibration isolation table can isolate the vibration transmission between the outside world and the measuring device, ensuring that the device is in a static state. In the static experiment, the self-designed measuring device is placed on the precision vibration isolation table steadily, and nine-axis data are collected for 45 min in a static state. The attitude is calculated using the gyroscope angular rate integration method and the proposed method, respectively. Due to the strong ferromagnetic interference in the vibration isolation platform environment, the static z-axis yaw angle is not calculated. The experiment results are shown in Figure 9.

It can be obtained from Figure 9 that due to the existence of the gyroscope zero bias, the error increases and diverges with the angular rate integral. The proposed attitude calculation method can effectively suppress the error caused by the angular rate integral calculation output by the gyroscope.

#### 4.2.3. Dynamic Hardware-in-the-Loop Simulation Experiment

In order to verify the dynamic performance of the proposed Kalman filter, a hardware-in-the-loop simulation platform was built according to Figure 10. A high-precision three-axis turntable was used for verification. By controlling the rotation of the inner frame and outer frame of the turntable, the pitch angle, roll angle, and yaw angle of the measuring device can be simulated. At the same time, the turntable controller can use the RS-422 serial port to realize synchronous position output.

Considering that the turntable is mainly composed of ferrous materials, the output of the magnetometer is greatly disturbed during the measurement process. Before the start of the test, the magnetometer ellipsoid fitting method based on the least square method was used to calibrate the output of the magnetometer. During the experiment, the measuring device was initially fixed in the center of the turntable and its XYZ axes were aligned with the NED Navigation Coordinate System. The experiment system was controlled to move quickly around the X and Z axes of the measuring device between 90° and −90°. In order to avoid the singularity problem in the pitch angle calculation process, the measuring device moves quickly between 80° and −80° in the *Y*-axis direction twice. The movement speed of the turntable is set to 50°/s, and the acceleration is set to 50°/s^2^. 

Using the solution method proposed in this paper, the extended Kalman (EKF), gradient-descent linear Kalman filter (GDLKF) and gradient-descent (GD) algorithm were used to analyze the experimental data. Where EKF uses the gyroscope output as the state vector and uses the accelerometer and magnetometer to directly calculate the attitude for a posteriori estimation, GDLKF uses gradient pose quaternions as observations for Kalman filtering. The experimental results are shown in Figure 11, Figure 12 and Figure 13.

It can be seen from Figure 11, Figure 12 and Figure 13 that the proposed method and EKF can accurately estimate the roll and pitch angles during the test. However, during the rotation of the turntable, the output of the accelerometer is affected by both the acceleration of gravity and the acceleration of the external motion, which result in a relatively large error when the motion state changes suddenly. Relatively, the proposed method in this paper has better dynamic performance than the GDLKF method and GD method.

Since the turntable is anisotropic iron equipment, it is subject to non-uniform magnetic interference during the rotation in the yaw angle calculation process. Although calibrated by the magnetometer, the four methods all produce errors. The proposed method update is significantly better than the other method. In order to more accurately reflect the discrepancy between different calculation methods, the root mean square error (RMSE) of the results of the simulation test and the hardware-in-the-loop simulation experiment is shown in Table 2. It can be seen from the table that the proposed method in this paper has better performance in a dynamic environment and can effectively track and estimate the attitude.

#### 4.2.4. Anti-Interference Experiment

In practical use, the environment is complex and there are many interferences. In order to verify the anti-interference performance and operational effect of the proposed attitude calculation method in a complex environment, the measurement device is fixed inside the test vehicle for practical application tests. The measurement device collects the MARG sensor system data and stores them in the data logger. After the experiment, the information in the data logger is read through the reserved interface, and then the calculation is performed in the PC. In this experiment, the third-generation Ellipse-N product of the French SBG company was used for measurement. The product has a built-in dual-frequency four-constellation GNSS module. Its integrated navigation output attitude angle is used as a reference value. The test environment and the vehicle trajectory recorded by GNSS are shown in Figure 14.

It can be seen from the satellite trajectory in Figure 14b that the experiment vehicle returns to the starting point after a week of driving, and the attitude of the measuring device is basically the same at the beginning and the end of the experiment, which can be used as an evaluation index for the attitude calculation effect. According to the calculation of the collected data, the result is shown in Figure 15, Figure 16 and Figure 17. The test results show that the root mean square error of the roll angle and the pitch angle is less than 0.7°, and the root mean square error of the yaw angle is about 1.3°.

## 5. Conclusions

In this paper, through the fusion of MARG sensor data, a new attitude calculation method combining a gradient descent algorithm and extended Kalman filter is proposed. The accelerometer and magnetometer data are processed through the two-stage gradient descent algorithm to correct the attitude angle, which effectively corrects gyroscope bias errors of the state vector. Meanwhile, compared with traditional external quaternion estimation methods, the proposed method can better eliminate the influence of magnetometer errors on the roll and pitch angles of the carrier. The proposed Kalman filter can provide relatively faster and more accurate attitude measurement results under different working conditions than using gradient descent and the linear Kalman filter alone.

## Figures and Tables

**Figure 1 micromachines-13-01283-f001:**
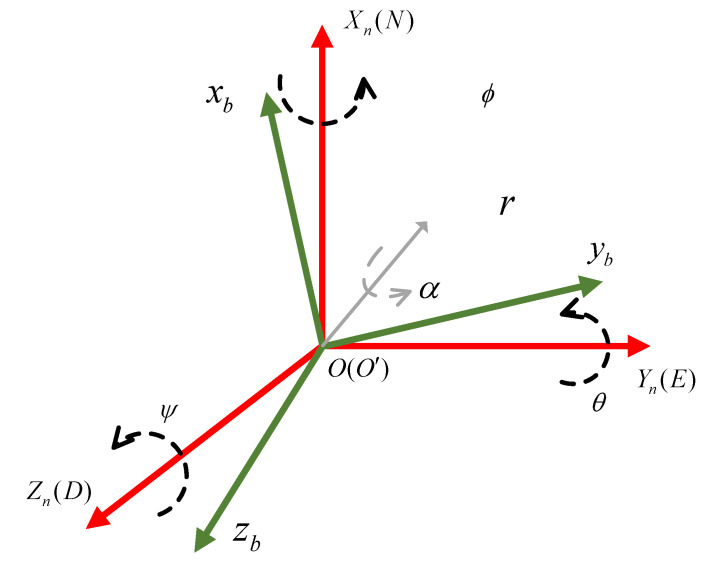
Carrier coordinate system and navigation coordinate system.

**Figure 2 micromachines-13-01283-f002:**
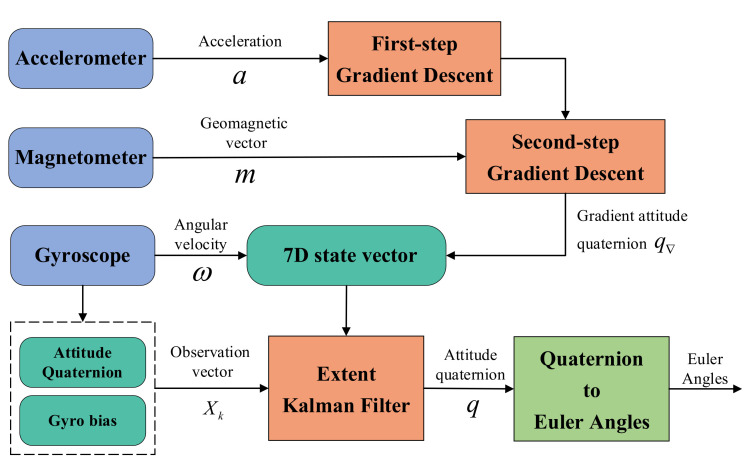
Block diagram of Kalman filter proposed in this paper.

**Figure 3 micromachines-13-01283-f003:**
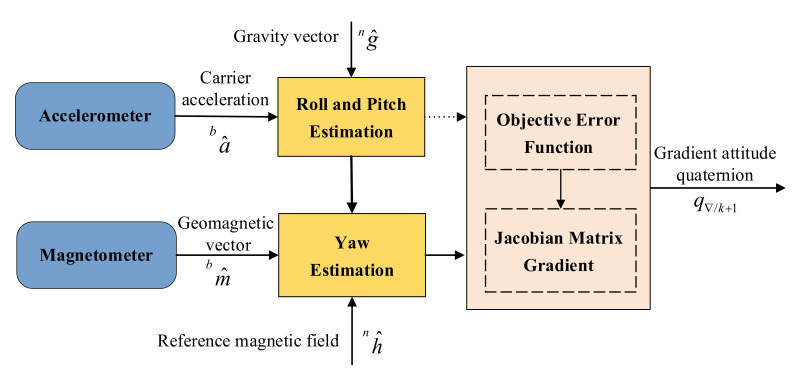
Schematic diagram of gradient descent algorithm.

**Figure 4 micromachines-13-01283-f004:**
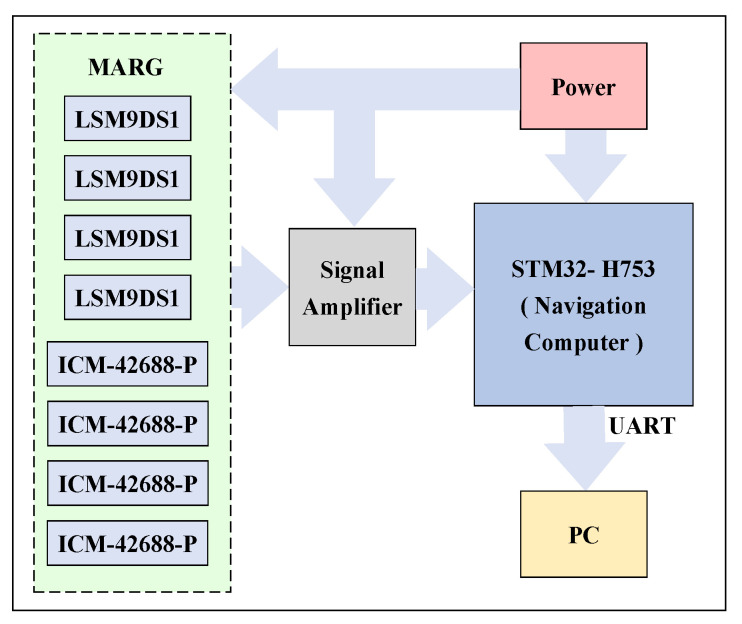
Block diagram of the hardware design.

**Figure 5 micromachines-13-01283-f005:**
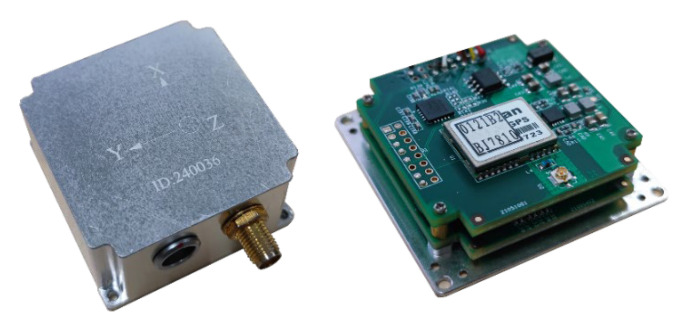
External and internal structure of physical design.

**Figure 6 micromachines-13-01283-f006:**
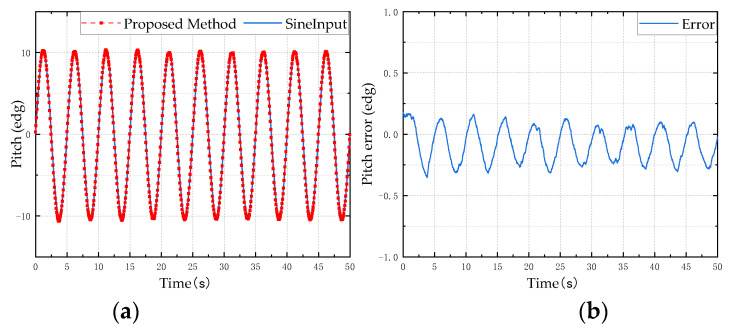
Pitch angle simulation test results: (**a**) is the estimation result of the proposed method; (**b**) is the error between the estimated value and the reference value.

**Figure 7 micromachines-13-01283-f007:**
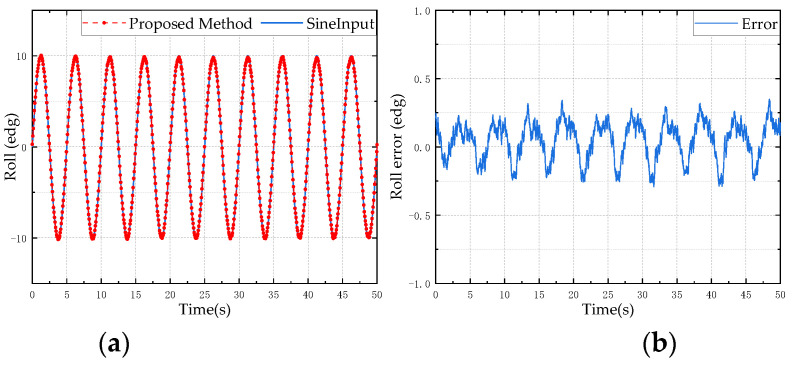
Roll angle simulation test results: (**a**) is the estimation result of the proposed method; (**b**) is the error between the estimated value and the reference value.

**Figure 8 micromachines-13-01283-f008:**
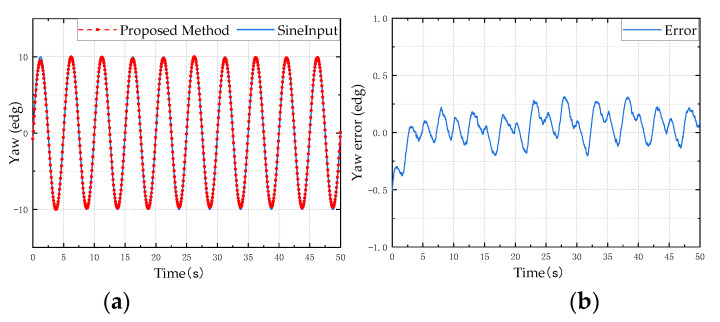
Yaw angle simulation test results: (**a**) is the estimation result of the proposed method; (**b**) is the error between the estimated value and the reference value.

**Figure 9 micromachines-13-01283-f009:**
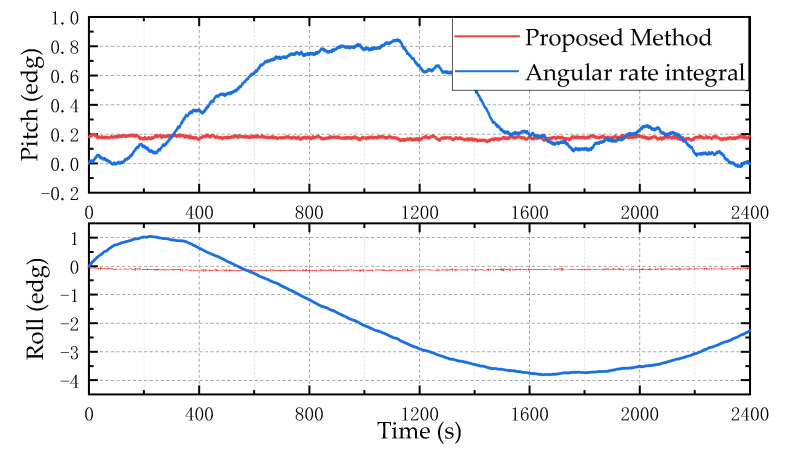
Static hardware-in-the-loop simulation test results. The top is the pitch angle result and the bottom is the roll angle result.

**Figure 10 micromachines-13-01283-f010:**
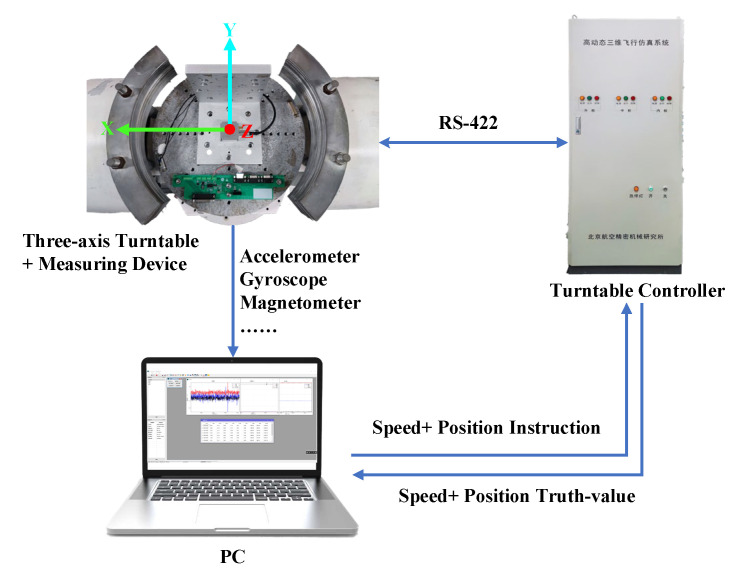
Schematic diagram of hardware-in-the-loop simulation.

**Figure 11 micromachines-13-01283-f011:**
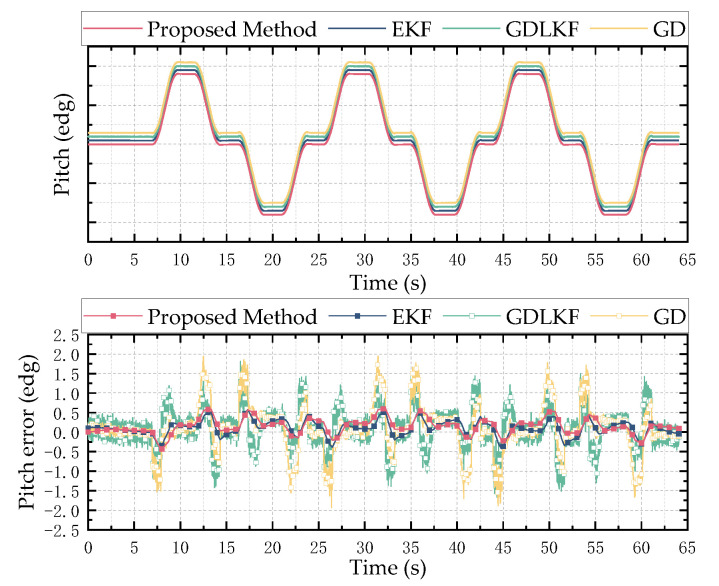
Hardware-in-the-loop simulation test results of pitch angle. The above is a comparison of the estimation results of different methods, and the bottom is the error between the estimated value of different methods and the reference value.

**Figure 12 micromachines-13-01283-f012:**
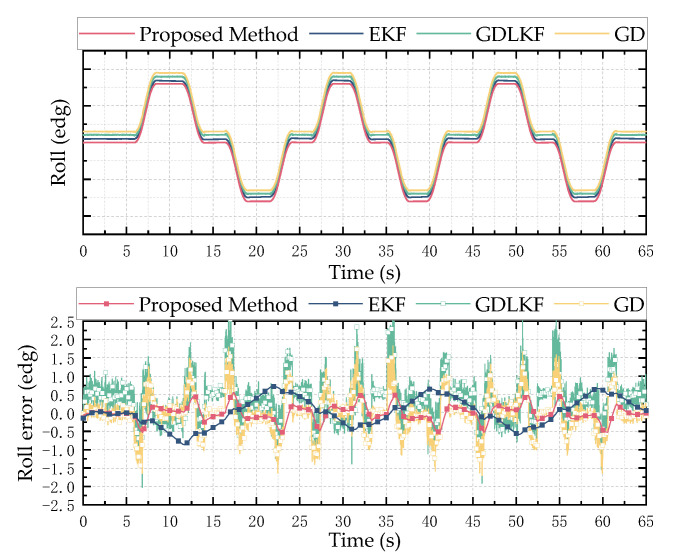
Hardware-in-the-loop simulation test results of roll angle. The above is a comparison of the estimation results of different methods, and the bottom is the error between the estimated value of different methods and the reference value.

**Figure 13 micromachines-13-01283-f013:**
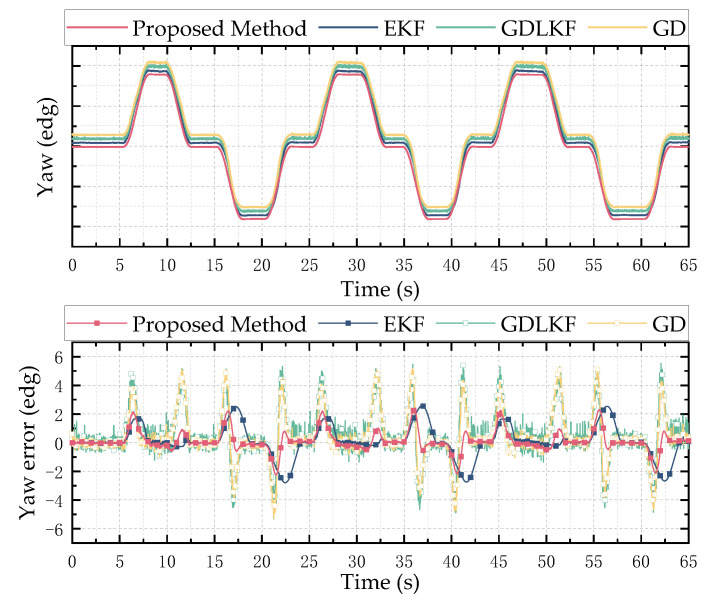
Hardware-in-the-loop simulation test results of yaw angle. The above is a comparison of the estimation results of different methods, and the bottom is the error between the estimated value of different methods and the reference value.

**Figure 14 micromachines-13-01283-f014:**
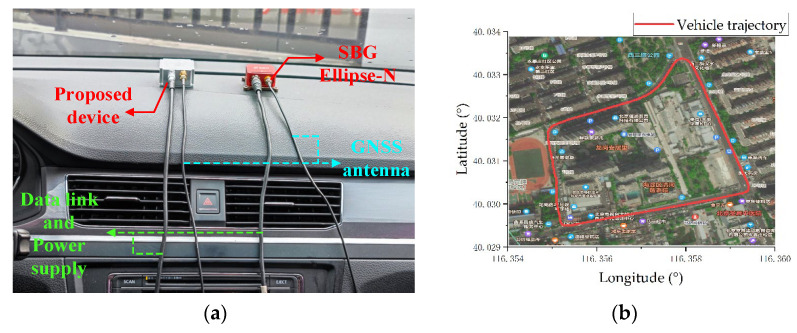
Anti-interference test equipment and environment: (**a**) is the experimental environment and experimental device; (**b**) is the trajectory of the vehicle.

**Figure 15 micromachines-13-01283-f015:**
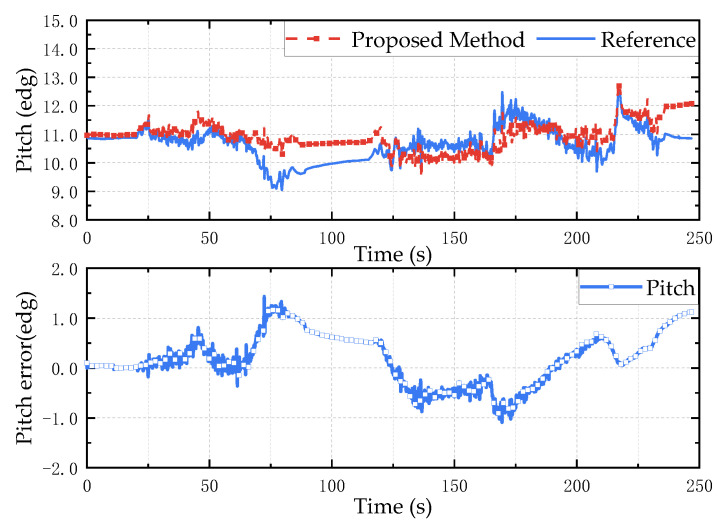
Pitch angle anti-interference test results. The above is the estimation result of the proposed method, and the bottom is the error between the estimated value and the reference value.

**Figure 16 micromachines-13-01283-f016:**
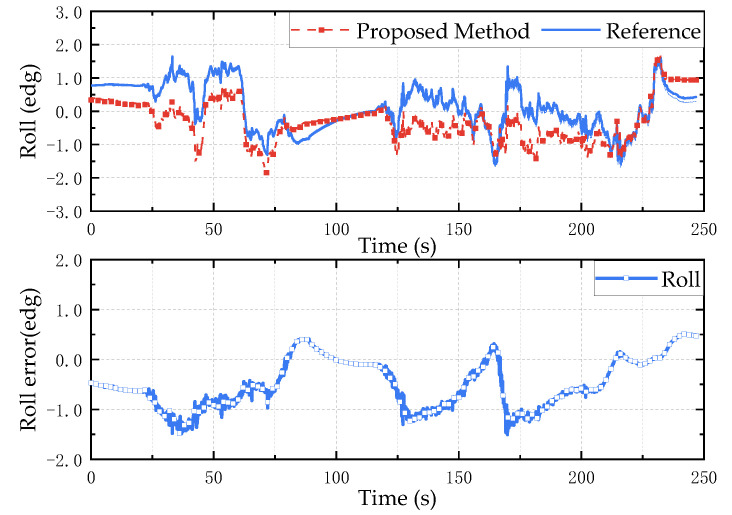
Roll angle anti-interference test results. The above is the estimation result of the proposed method, and the bottom is the error between the estimated value and the reference value.

**Figure 17 micromachines-13-01283-f017:**
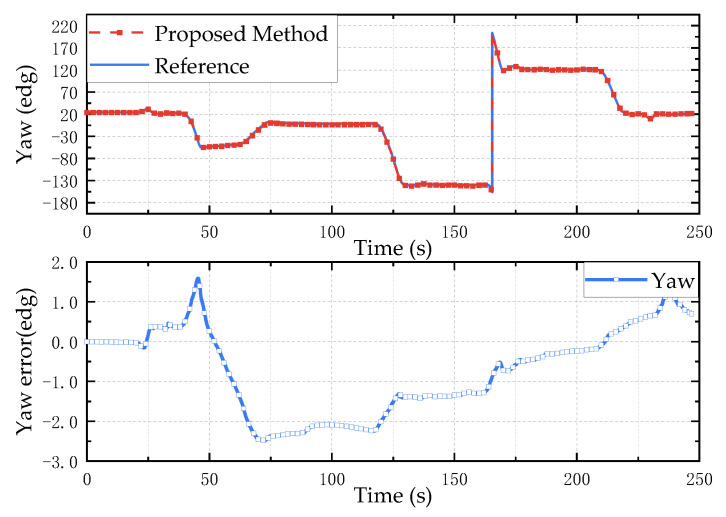
Yaw angle anti-interference test results. The above is the estimation result of the proposed method, and the bottom is the error between the estimated value and the reference value.

**Table 1 micromachines-13-01283-t001:** The parameters of the MARG sensor of the self-designed measuring device.

Sensor	Bias	Standard Deviation
Gyroscope	0.2°/s	0.05°/s
Accelerometer	±5 mg	0.0055 mg
Magnetometer	±1 mGauss	0.1 mGauss

**Table 2 micromachines-13-01283-t002:** Root mean square error of simulation experiment and semi-physical simulation experiment.

Experiment Type	Algorithm	Pitch/°	Roll/°	Yaw/°
Simulation experiment	Proposed Method	0.3330	0.3099	0.4051
	EKF	0.4501	0.2975	0.8899
	GDLKF	0.3430	0.3313	1.1408
	GD	0.1056	0.1790	0.4177
Semi-physical simulation experiment	Proposed Method	0.2072	0.2169	2.5589
	EKF	0.6087	0.4976	2.6848
	GDLKF	0.4976	0.6087	2.6643
	GD	0.2169	0.3108	2.6093

## Data Availability

Publicly available datasets were analyzed in this study. This data can be found here: [https://gitee.com/bistu_liuning/car-navigation-data, accessed on 22 August 2021].

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
