# Peer review of "Research on Gradient-Descent Extended Kalman Attitude Estimation Method for Low-Cost MARG"

_micromachines, 2022, doi:10.3390/mi13081283_

Round 1
Reviewer 1 Report
This paper aims to improve the accuracy of attitude estimation by fusing MARG sensor information, and proposes an attitude estimation method combining Extended Kalman Filter and gradient quaternion method. In particular, the article introduces compensation for gyroscope errors. The MARG sensor system is designed to verify the performance of the filter. The results show that it has good dynamic performance and measurement accuracy. The paper is well written, with detailed derivations and through reference revision of the applied methods. However those following problems existed:
1) In the keywords list, the authors are recommended to use "attitude estimation" consistent with the title, instead of "attitude measurement". Moreover, "quaternion" is not a good keyword, it is recommended to modify or delete.
2) Some details: line179, 194, 198, 333, The format of the numbers is obviously inconsistent with the text; line 190,306 also has some obvious format errors.
3) The symbol of the attitude angle in line 116 is inconsistent with the representation in the figure.
4) There are some equations whose symbols are not clearly stated, eg. equations35,38,39,44. In addition, The gyroscope attitude quaternion in equations 11 should be distinguished from the previous one to avoid ambiguity.
5) What is the unit of Table(2)? RMSE should be the same as the original data unit.
6) What was the framework of the EKF, which was used in the comparison. It should be introduced in the text.
7) The references list needs to be updated. There are old references that can be replaced with more recent ones.
Reviewer 2 Report
Research on Gradient Descent Extended Kalman Attitude Estimation Method for Low-cost MARG
In this manuscript, the authors propose an Extended Kalman Filter suitable for low-cost Magnetic, Angular rate, and Gravity (MARG) sensor system. This is a combination of two-stage gradient descent algorithm and Extended Kalman Filter (GDEKF). First, the accelerometer and magnetometer are used to correct the attitude angle according to the two-stage gradient descent algorithm. The obtained attitude quaternion is combined with the gyroscope measurement value as the observation vector of EKF and the calculated attitude of the gyroscope and the bias of the gyroscope are corrected. The elimination of the bias of the gyroscope can further improve the stability of the attitude observation results. Finally, the MARG sensor system was designed for mathematical model simulation and hardware-in-the-loop simulation to verify the performance of the filter. The results show that compared with the gradient descent method, it has better anti-interference performance and dynamic performance, and better measurement accuracy than Extended Kalman Filter.
The article is well presented and the topic interesting. I can recommend this for publication after the next suggestions are considered:
§ In the introduction, I recommend expanding the list of potential studies with a particular interest in the integration between positional technologies. For example, the integration between inertial systems/units (INS/IMU) and GPS. For example, this topic is very relevant in studies related to traffic psychology and driving behavior. Some studies apply naturalistic methodologies (Naturalistic driving) for studying the driving behavior in actual driving conditions based on large-time observations and monitoring of a great number of indicators. However, quite often many studies refer to the problems with positional data at some point due to unmatching between technologies/sensors and this leads to inconsistencies with the rest of the measurements. I can suggest some papers related to the problem with this such as “Geo-referencing naturalistic driving data using a novel method based on vehicle speed” and “Preprocessing of data for recovery of positioning data in naturalistic driving trial”. The same author refers to the problem of the integration between INS and GPS for naturalistic driving studies with road vehicles. Other potential studies where your topic is very significant are for airborne laser scanning and aerial mapping, where the position of the aircraft has to be very accurately defined every time.
§ Line 17-> delete the “.” in “EKF.”.
